# Estimation of Mean Radiant Temperature in Urban Canyons Using Google Street View: A Case Study on Seoul

**Eun-Sub Kim [1], Seok-Hwan Yun [1]**(ID), **Chae-Yeon Park [2]**(ID), **Han-Kyul Heo [3] and Dong-Kun Lee [1,4,]***(ID)

[1] Interdisciplinary Program in Landscape Architecture, Seoul National University, Seoul 08826, Korea; mr.solver92@snu.ac.kr (E.-S.K.); ysw330616@snu.ac.kr (S.-H.Y.)

[2] Center for Social and Environmental Systems Research, National Institute for Environmental Studies, Tsukuba 305-0053, Japan; chaeyeon528@snu.ac.kr

[3] Architecture & Urban Research Institute, Sejong 30103, Korea; hkheo@auri.re.kr

[4] Smart City Global Convergence Program, Seoul National University, Seoul 08826, Korea

* Correspondence: dklee7@snu.ac.kr; Tel.: +82-10-3227-5435

**Abstract:** Extreme heat exposure has severe negative impacts on humans, and the issue is exacerbated by climate change. Estimating spatial heat stress such as mean radiant temperature (MRT) is currently difficult to apply at city scale. This study constructed a method for estimating the MRT of street canyons using Google Street View (GSV) images and investigated its large-scale spatial patterns at street level. We used image segmentation using deep learning to calculate the view factor (VF) and project panorama into fisheye images. We calculated sun paths to estimate MRT using panorama images from Google Street View. This paper shows that regression analysis can be used to validate between estimated short-wave, long-wave radiation and the measurement data at seven field measurements in the clear-sky (0.97 and 0.77, respectively). Additionally, we compared the calculated MRT and land surface temperature (LST) from Landsat 8 on a city scale. As a result of investigating spatial patterns of MRT in Seoul, South Korea, we found that a high MRT of street canyons (>59.4 °C) is mainly distributed in open space areas and compact low-rise density buildings where the sky view factor is 0.6–1.0 and the building view factor (BVF) is 0.35–0.5, or west-east oriented street canyons with an SVF of 0.3–0.55. However, high-density buildings (BVF: 0.4–0.6) or high-density tree areas (Tree View Factor, TVF: 0.6–0.99) showed low MRT (<47.6). The mapped MRT results had a similar spatial distribution to the LST; however, the MRT was lower than the LST in low tree density or low-rise high-density building areas. The method proposed in this study is suitable for a complex urban environment consisting of buildings, trees, and streets. This will help decision makers understand spatial patterns of heat stress at the street level.

**Keywords:** Street canyon; Google Street View; deep learning; mean radiant temperature; thermal comfort

## 1. Introduction

Exposure to heat may cause severe illnesses and deaths during intense heat events, especially in large urban areas, because of altered urban climate conditions [1–3]. As future climate change scenarios of heat-related diseases and mortality have become a major public health issue, a higher spatiotemporal assessment map of heat stress in cities is needed [4–8]. Accordingly, several studies have used satellite images to retrieve the land surface temperature (LST) in large cities, which evaluated the urban thermal environment and determined the relationship between microclimate conditions and human health-related heat stress or urban geometry [9–12]. Concordance between the LST-derived satellite and ambient temperatures measured showed a similar tendency in certain atmospheric conditions [13]. Therefore, LST-derived from satellite imagery provides results for establishing local measurements of heat exposure. However, satellites are insufficient for estimating heat stress in

local areas because their coarse spatial resolution, and obstacles such as clouds, cost, and finer temporal resolution, remain significant limitations [14].

The mean radiant temperature (MRT), which is the amount of heat received by the urban environment in the city at a specific location, is an effective heat-related health impact indicator in urban environments [3]. MRT is an important daytime human bio-meteorological variable on no cloud days [14,15] and it is highly correlated with heat-related diseases and heat stress [16].

Many models have been developed to evaluate the MRT in urban canyons. For example, Perini et al. [17] increased nighttime outdoor thermal comfort accuracy by incorporating ENVI-met and transient systems simulation (TRNSYS). Perini et al. [17] evaluated the MRT of Shanghai city during the summer using a geographic information system (GIS)-based simulation approach, the SOLWEIG model, to reflect urban three-dimensional constructions. Two studies [18,19] calculated the radiant flux and shadow spaces using weather data and other input data at the point of location to estimate the MRT. In particular, the MRT value for all-sky can be derived using the formula proposed by Matzarakis et al. [19] in the Rayman model. However, previously developed models require complicated urban-structure input data for calculating the sky view factor (SVF) and radiative data, such as shortwave radiation fluxes, to evaluate at the street level [20,21]. The MRT formula is complex and difficult to evaluate on a city scale [22]. Therefore, we suggest a method that can estimate the MRT using a simplified equation and input data at the city scale.

Panorama images acquired through Google Street View (GSV) provide public and freely accessible data. GSV was developed to analyze street geometric characteristics (buildings, trees, and the sky) through image classification algorithms using deep learning and converting panorama images into fish-eye images [23]. In other words, panorama images can be a useful tool for estimating shortwave radiation at the street level on a large city scale [23–25]. In particular, the image projected from the panoramic image to the cylindrical can estimate incoming shortwave radiation at a particular point, with accurate view factors (from the sky, buildings, and trees) using deep learning [26,27]. Although the method of calculating shortwave radiation was described in a previous study, there has been no attempt to calculate MRT using radiation fluxes from the view factor (VF).

As a result, we proposed a new method to estimate city-scale MRT using street geometrics characterized with panorama images. The aim of this study was to: (1) present a street-level MRT estimation method with image segmentation using deep learning; (2) validate the calculated MRT with two types of observation-based data, i.e., MRT and LST measured from Landsat 8; and (3) identify which street geometric factors affect the spatial variation in MRT. The proposed method can calculate high-resolution MRT over a wide spatial range by using freely obtained panorama images, which will help heat-related health studies and thermal-friendly urban design and planning.

## 2. Study Area and Data Collection

### 2.1. Study Area

Seoul, South Korea is one of the largest and most densely populated cities in the world (Figure 1). The city has a population of approximately 9.8 million living in ~606 km$^2$ of developed land. Situated west of the central part of Korea with a basin surrounded by mountains, Seoul is cold in the winter because of the dry continental high pressure, and hot and humid in the summer because of the high temperature and humidity in the North Pacific Ocean.

Seoul has experienced rapid urbanization with high-density buildings and associated urban heat island (UHI) effects. UHI significantly affects disease-related heat stress living conditions. In addition, Seoul is a metropolitan area with the highest population density among OECD member countries the elderly population accounts for 9.3% of the total population [28]. Therefore, the number of people vulnerable to heat is high and predicted to increase as a result of a continued increase in the elderly population ratio and urbanization.

Seoul was selected as a representative highly urbanized area in this study to estimate the spatiotemporal MRT was evaluated using panorama images.

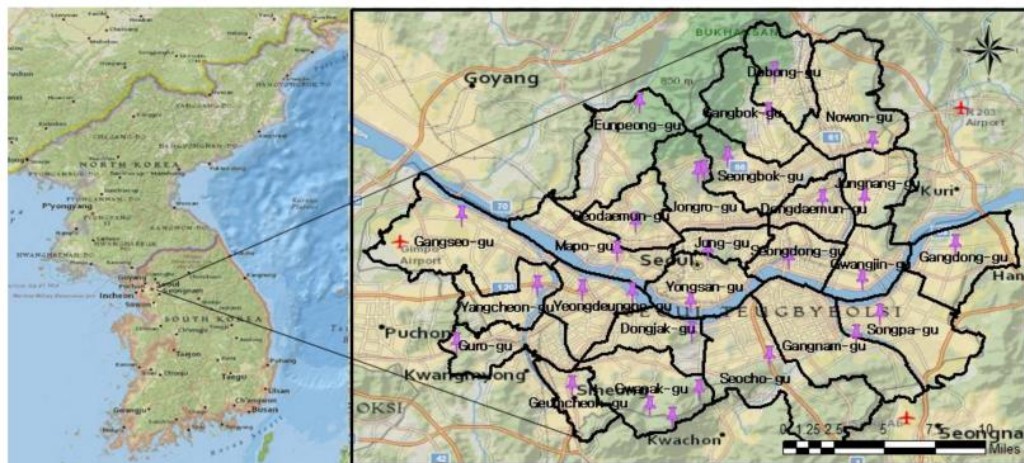

**Figure 1.** Location of the study area (City of Seoul). The total area is about 605.21 km$^2$ and the population of the city is 9.8 million.

### 2.2. Data Collection Area

We collected three datasets (panorama images acquired through Google Street View (GSV), meteorological data, landsat8 satellite images) to estimate MRT and compare LST and MRT. We collected a total of 344,044 street panorama images sampled using the GSV application programming interface (API) at 30-m intervals in the high-density urban area of Seoul Data in summer. To derive the solar path in the fisheye image of a street canyon, we adjusted panorama images to the north by shifting the vehicle direction [24]. We also collected air temperature, humidity, shortwave radiation, and longwave radiation using net radiometers CNR4 [29] and S-thb0m002 equipment to validate our method (see Appendix A). Meteorological data (air temperature and humidity) were collected from a government website of an online resource (https://data.kma.go.kr accessed on 9 September 2021) and used for point data within 4 km of the AWS location to estimate the MRT (Table 1).

**Table 1.** Study area for verification and application in each 2018–2020. Latitude (Lat.) and Longitude (Log.) were used for collecting panorama images.

| | Object | Location | Lat. | Log. | Data Collection Date | Input Data | No. of Panorama Images |
|---|---|---|---|---|---|---|---|
| Google Street View | Validation sites | Low building | 37.457391 | 126.948493 | 18.04.16~18 | Radiation (shortwave, longwave), air temperature, relative humidity | 7 |
| | | Park | 37.495193 | 127.003546 | 18.04.19~21 | | |
| | | Commercial area | 37.521532 | 126.927314 | 18.04.28~30 | | |
| | | Apartment | 37.503094 | 126.943548 | 18.05.04~07 | | |
| | | River | 37.528474 | 126.934370 | 18.05.10~13 | | |
| | | Narrow alley | 37.482026 | 126.929579 | 18.05.31~06.03 | | |
| | | Residential area | 37.469727 | 126.942584 | 18.06.01~04 | | |
| | Mapping | Seoul | 37.34~37.5666 | 126.584~126.978 | 2014~2020 (4~10) | Air temperature Relative humidity | 58,794 |
| Satellite image | Attribute | | | | | | |
| | Date | Satellite image | Projection | Datum | Cloud cover (in %) | Sensor | Time |
| | 2018. 06. 19 | Landsat 8 | UTM zone52 | WGS84 | 1.79 | OLI_TIRS | 09:52 |

Finally, we collected Landsat 8 OLI/TIRS C2 L2 images to analyze the differences between LST and MRT. Two Landsat 8 OLI/TIRS images covering Seoul with <10% cloud cover were selected. For Landsat 8 data, bands 4, 5, and 10 were used to derive the LST. The Landsat8 data set is currently available free of charge from the USGS website (https://earthexplorer.usgs.gov accessed on 9 September 2021). Satellite image acquired on 19 Jun 2018 (OLI & TIRS).

## 3. Method

This manuscript presents an estimation method of MRT with an image segmentation technique using deep learning in high-density urban areas and analyzes the comparison between LST and MRT on a city scale. To accurately estimate the MRT, the VF and surface temperature must be accurately calculated. However, calculating the VF and surface temperature from each urban element in a street canyon is difficult.

First, we implemented the scene parsing method in a deep-learning framework [30]. Then, we calculated the longwave radiation using the rate at which shortwave radiation is absorbed by the urban surface based on the measured data [31,32]. To verify the accuracy of the method presented in this study, we compared the MRT of seven field measurements. Finally, we comparatively analyzed the MRT and LST in Seoul and identified the geometric characteristics of high MRT and LST.

### 3.1. Schematic Framework

The schematic framework for this study is presented in Figure 2 and consists of three main phases. In phase I (Figure 2, blue), based on the image classification method using deep learning, urban structures such as skies, trees, and buildings were extracted. The classified image was used when calculating the VF in Section 3.2. The image segmentation using deep learning with the panorama image classified the complex urban environment for each object, and quickly derived the View Factor (VF) and Sky View Factor (SVF) using deep learning (see Appendix B). In particular, a panorama image in an urban canyon is a useful tool to create urban structure data. The panorama to fisheye image projection method and sun path algorithm proposed in Gong et al. [23] and Reda and Nrel [33]. detected the presence or absence of shadows during the day. In Phase II (green rectangle), the shortwave and longwave radiations were calculated for MRT calculation at the street level under a clear sky. Short wave radiation was calculated for each SVF and VF estimated with image segmentation using deep learning. In particular, longwave radiation was estimated by using the ratio method of converting shortwave radiation to longwave radiation based on the results of 50 years of data [31,34,35]. Then, combined with street morphologies and the solar path to derive the clear-sky street-level solar radiance, in Phase III (yellow rectangle), the MRT was estimated by summing the shortwave and longwave radiation calculated using direct, diffuse, and reflected radiation fluxes.

### 3.2. View Factor Calculation and Shadow Detection

Images were initially classified using the scene parsing method in deep learning based on previous research [36,37]. In this study, the VF for each urban element (buildings, trees, sidewalks, roads) was calculated using a method from a previous study [27] and the albedo, absorption coefficient, and heating coefficient were assigned (Table 2). View factor (VF) is used in the equations for calculating the reflected radiation in Sections 3.3.1 and 3.3.2).

We then projected the panorama images from cylindrical to azimuthal projection to generate fisheye images using the photographic method by matching the pixels on fisheye and panorama images [24] as shown in Figure 3. The solar zenith and azimuth angles were calculated using input data (date, time, latitude, and longitude) from the collected panorama images to calculate the sun path during the day. Then, the shadow was calculated by overlaying the sun path coordinates onto the fisheye image.

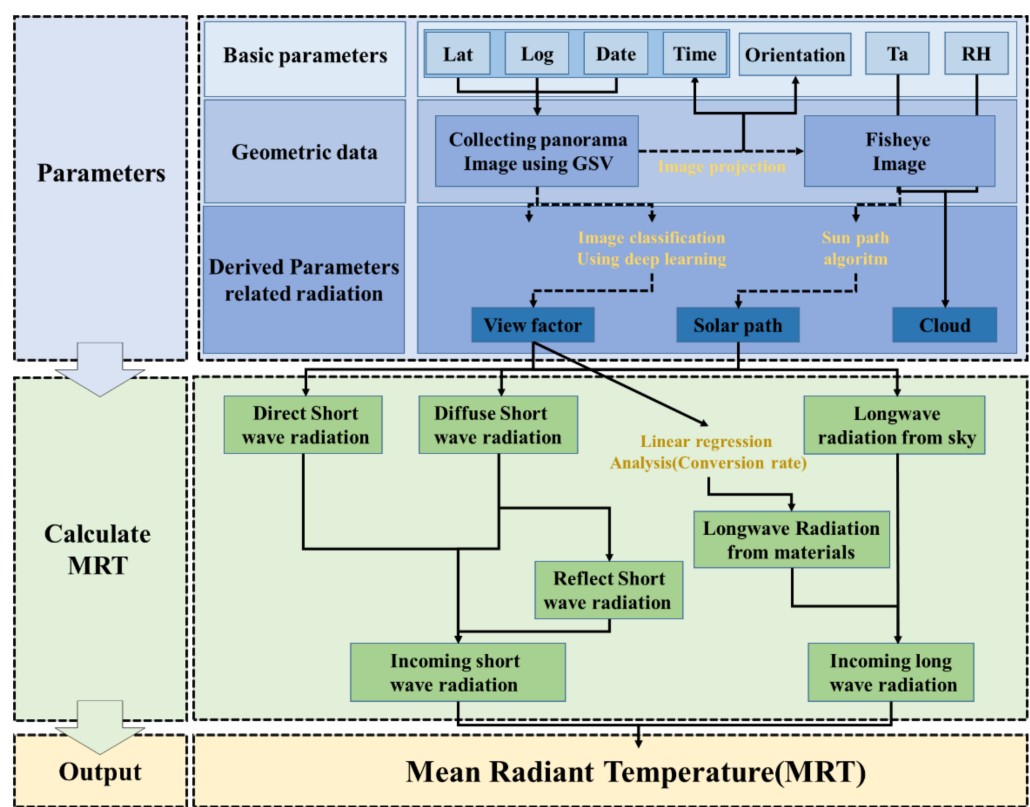

**Figure 2.** Schematic flow for calculating MRT using deep learning.

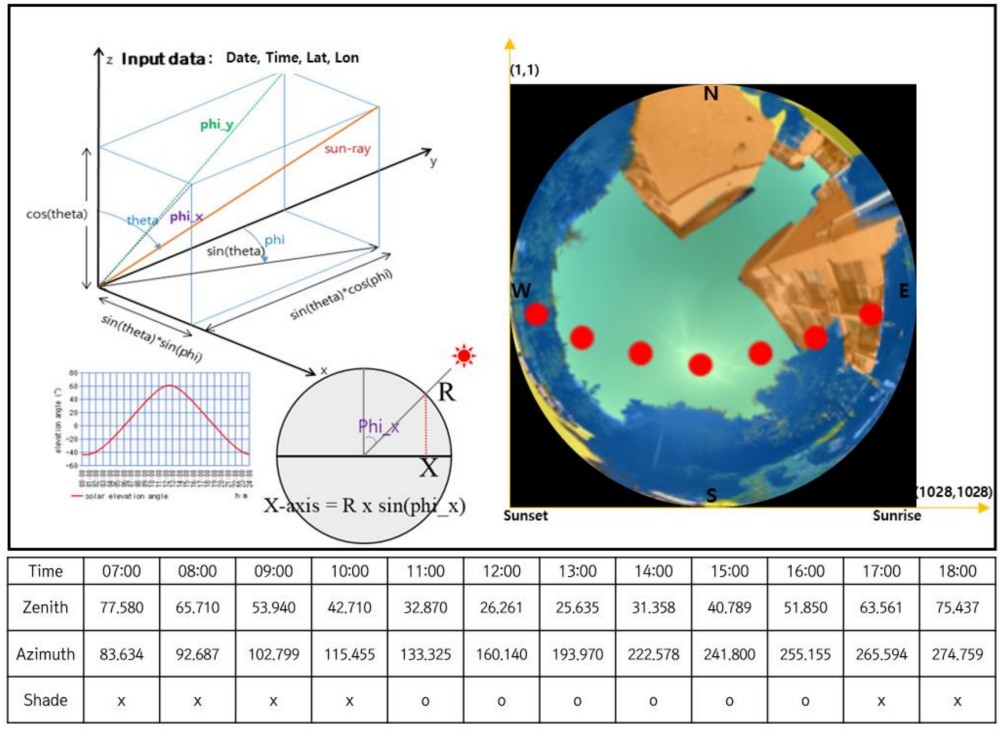

| Time | 07:00 | 08:00 | 09:00 | 10:00 | 11:00 | 12:00 | 13:00 | 14:00 | 15:00 | 16:00 | 17:00 | 18:00 |
|---|---|---|---|---|---|---|---|---|---|---|---|---|
| Zenith | 77.580 | 65.710 | 53.940 | 42.710 | 32.870 | 26.261 | 25.635 | 31.358 | 40.789 | 51.850 | 63.561 | 75.437 |
| Azimuth | 83.634 | 92.687 | 102.799 | 115.455 | 133.325 | 160.140 | 193.970 | 222.578 | 241.800 | 255.155 | 265.594 | 274.759 |
| Shade | x | x | x | x | o | o | o | o | o | o | x | x |

**Figure 3.** Solar path in fisheye image using input data (date, time, latitude, longitude). Red point shows the projected sun positions for 2020.08.18 on hemispherical images. For a person at the location of (126.9658, 37.5714) and time (t), when the red point (sun position) is located on the sky pixel in the hemispherical images, the person is exposed to direct sunlight. If the sun position is on non-sky pixels, the person at that time is shaded from sunlight.

**Table 2.** Materials table presenting absorption coefficient, albedo, heating coefficient.

| Material | Absorption Coefficient | Albedo | Heating Coefficient | Reference |
|---|---|---|---|---|
| Building | 0.6 | 0.4 | 0.08 | |
| Pavement | 0.86 | 0.14 | 0.08 | Park et al. [15]; Offerle et al. [32] |
| Sidewalk | 0.7 | 0.3 | 0.08 | |
| Tree | 0.85 | 0.15 | 0.25 | Barad et al. [35] |
| Grass | 0.75 | 0.25 | 0.25 | Holtslag and Ulden [31] |
| Soil | 0.7 | 0.3 | 0.38 | Barad et al. [35] |

*3.3. Calculation of Total Shortwave Radiation in Street Canyon*

3.3.1. Calculation of Street-Level Shortwave Radiation

The FAO showed high correspondence with the measured clear-sky radiation ($I_g$) data based on an estimated 0.75 extraterrestrial radiation ($R_a$). To calculate clear-sky shortwave radiation, the angstrom formula [38,39], which relates shortwave radiation to extraterrestrial radiation, was used:

$$I_g = 0.75R_a \left(1 - 0.75^{FCLD}\right)^{3.4} \tag{1}$$

$$I_g = R_a \left(0.75 + 2 \times 10^{-5} \times z\right) \tag{2}$$

Equation (1) reflects the shadow and can be derived from the shortwave radiation from the actual daily duration of sunlight in hours per day. According to the Korea Meteorological Administration, the height of the densely populated areas in Seoul is 15 to 60 m above sea level, and then substituting this into z gives 0.7503 to 0.7512 (Equation (2)). In addition, when the height of high-rise buildings (200 m) is substituted, the value is 0.754. Therefore, the value of 0.75 in Equation (2) can be estimated by considering shadow radiation in urban areas with high-density buildings.

To estimate the shortwave radiation decrease due to cloud coverage as soon as it enters the atmosphere, (1) the cloud cover was calculated using air temperature and humidity data, and (2) direct radiation ($S_{dir}$) and diffuse radiation ($S_{diff}$) were separated using the cloud radiation model (CRM) developed by [40,41] to obtain a better assumption under more complex conditions.

$$S_{diff} = I_g \left(0.3 + 0.7 \times FCLD^2\right) \tag{3}$$

$$S_{dir} = I_g - S_{diff} \tag{4}$$

In this study, shortwave radiation was calculated using the models of Allen et al. [39] as shown in the above formulas in Equations (1)—(4). The direct incident ($D_{dir}$), scattering ($D_{diff}$), and reflected shortwave radiation ($D_{reflect}$) were calculated using the following equations:

$$D_{dir} = S_{dir} \times f \tag{5}$$

$$D_{diff} = S_{diff} \times \Psi_{sky} \tag{6}$$

$$D_{reflect} = \sum_{n=1}^{k} (D_{dir} + D_{diff}) \times a \times VF_{materials} \times f \tag{7}$$

where $f$ is a binary value (0,1) according to the presence or absence of a shadow at the measurement point. When $f = 0$, the ray path is masked by an obstacle. $k$ is the number of materials of the classified image; in this study, it was set to 6 and is the albedo for each material.

### 3.3.2. Calculation of Street-Level Longwave Radiation

The steps for calculating longwave radiation were as follows: (1) incoming longwave radiation from the atmosphere ($L_\downarrow$); (2) emitted longwave radiation from urban elements ($L_\rightarrow$); and (3) outgoing longwave radiation ($L_\uparrow$) were calculated using the net all-wave radiation parameterization (NARP) model proposed by Offerle et al. [32]. The NARP model calculates the longwave radiation by considering the amount of water vapor in the atmosphere and the cloud cover (Equation (6)). The longwave radiation process of this study is more specific than the previously developed method [39], and the following formula was used:

$$L_\downarrow = (\epsilon_{clear} + (1 - \epsilon_{clear})FCLD)\sigma T^4 \tag{8}$$

As mentioned above, the incoming longwave radiation from the atmosphere was calculated using the NARP model [32] and the SFV estimated from the fisheye image.

$$L_{sky} = L_\downarrow \times \Psi_{sky} \tag{9}$$

The incoming longwave radiation from urban elements in urban canyons was calculated for all VFs except SVF, using the longwave radiation conversion ratio (heating coefficient, Table 2). In previous studies, in order to accurately estimate the emitted longwave radiation, the heat coefficient for each material was derived by conducting a regression analysis between $4\sigma T^3(T_s - T)$ and net radiation through measurement data. However, the heat coefficient ($Fsw$) has a large range depending on the materials. In this study, 0.25 values were used for grass and trees, and 0.38 values were used for bare ground cover, referring to the results derived from previous studies [35]. And for other materials, 0.08 was used to estimate longwave radiation [31,32,35] (Table 2).

$$L_\rightarrow = \left(VF_{materials} \times \left(L_\downarrow + Fsw \times \left(S_{dir\downarrow} + S_{diff\downarrow}\right)\right)\right) \times VF_{materials} \tag{10}$$

where $VF_{materials}$ is the view factor for each urban materials. In this study, $VF_{materials}$ is both BVF and TVF. Outgoing longwave radiation is the amount of radiation emitted from storage heat surfaces by the amount of shortwave radiation and was calculated using the heating coefficient and (1-albedo). In this study, the atmospheric temperature was used to estimate longwave radiation. Simple outgoing long-wave radiation formula were derived using Equations (11) and (12). Equation (13) is calculated through the atmospheric temperature replacing the surface temperature and the estimated short-wave radiation.

$$\varepsilon_0\sigma T_0^4 \approx \varepsilon_0\sigma T_a^4 + 4\varepsilon_0\sigma T_a^3(T_0 - T_a) \tag{11}$$

$$4\varepsilon_0\sigma T_a^3(T_0 - T_a) = FswK_\downarrow(1 - a_0) \tag{12}$$

$$L_\uparrow = \sigma T^4 + Fsw\left(S_{dir} + S_{diff}\right) \times (1 - a) \tag{13}$$

### 3.3.3. Calculate Mean Radiation Temperature

The MRT was calculated according to the radiation from the urban elements surrounding a pedestrian [42]. These elements reflect direct and diffuse shortwave radiation and emit and reflect longwave radiation, which are summed using Equation (12) from [19].

$$MRT = \left(\frac{1}{\sigma}\sum_{i=1}^{n}\left(E_i + a_k\frac{D_i}{\epsilon_p}\right)\right)^{0.25} - 273.15 \tag{14}$$

where $a_k$ is the absorption coefficient and $\epsilon_p$ is the emissivity of the pedestrian, which have standard values of 0.7 and 0.97, respectively [41].

## 4. Results and Discussion

### 4.1. Verification of Shortwave Radiation Estimated at Street-Level

#### 4.1.1. Validation of Shortwave Radiation

Figure 4 shows a consistent variation trend in the validation between the measured data and the estimated shortwave and longwave radiation. The estimated shortwave radiation was slightly higher than the measured data. The calculated and measured shortwave radiation values were between $-19$ and $+23$ W/m$^2$ during the daytime.

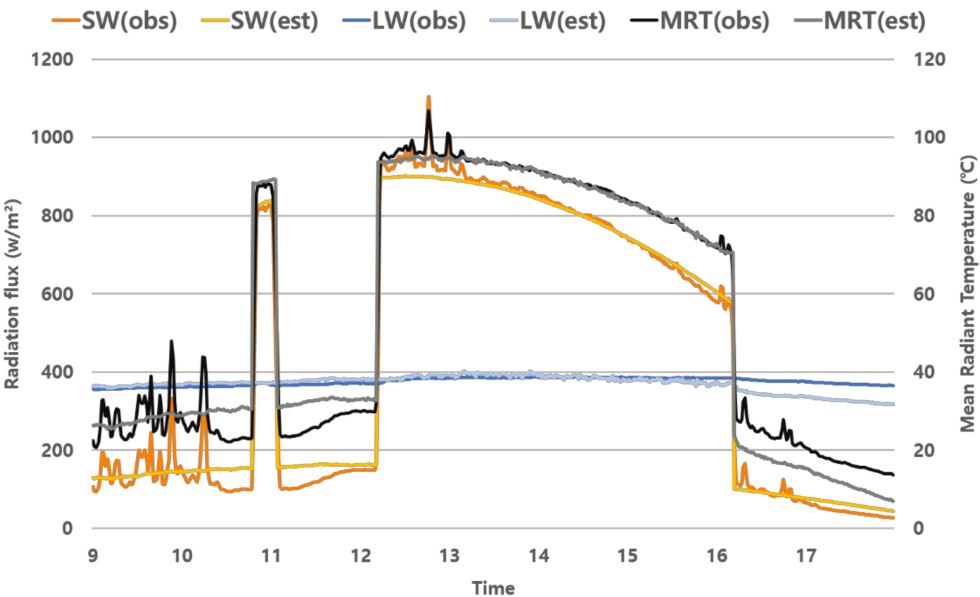

**Figure 4.** Comparison between estimated shortwave, longwave radiation, MRT and data measured in Korea over hourly. Orange, yellow line is incoming shortwave radiation, blue line is incoming longwave radiation, black, gray line is MRT.

This may be due to the patterns of anthropogenic energy usage which decrease storage efficiency at high wind speeds and overestimated SVF. However, Figure 4 show that the shadow detection using the sun path algorithm is captured well from the changing shortwave radiation. In the comparison between the estimated longwave radiation and the measured data, the difference is larger than that of the shortwave radiation (Figure 5). In particular, Appendix. C (b, d, f, h, j) shows that the differences between the outgoing longwave radiation is $-20$ W/m$^2$ and $+50$ W/m$^2$, respectively, between 12 and 2 pm. The incoming longwave radiation was underestimated during sunset. The cause of overestimation and underestimation of longwave radiation is the inaccurate surface temperature. Sites with many impermeable surfaces such as buildings, sidewalks, and pavements, were overestimated, and sites with a high percentage of green space were underestimated. To accurately estimate longwave radiation, the Fsw ratio must be measured at the point of estimation.

The method proposed in this study for estimating shortwave and longwave radiation was generally very well correlated $\left(R^2 = 0.98, R^2 = 0.77\right)$ (Figure 5). It means that shortwave radiation can detect the presence of shadows well. Although formulas of longwave radiation estimated by using the ratio method of converting shortwave radiation to longwave radiation requires simplification, estimated longwave radiation is tightly correlated. Therefore, we can conclude that the spatial pattern of the shortwave radiation at street level can be predicted by the estimated MRT with the image segmentation technique using deep learning.

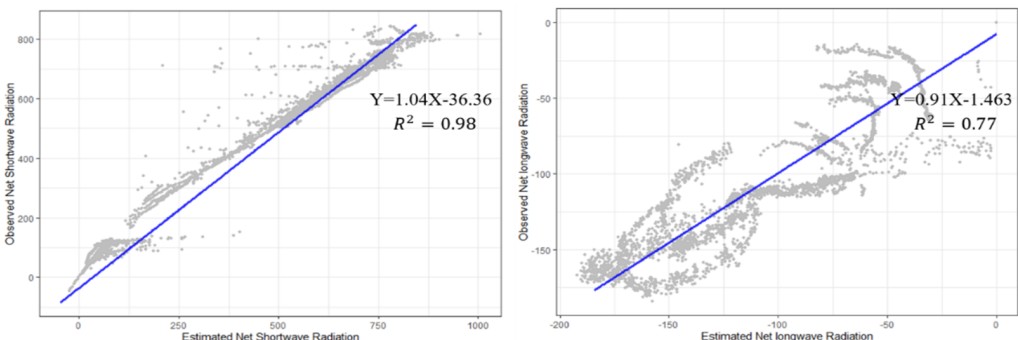

**Figure 5.** Validation of Shortwave radiation $R^2 = 0.98$ and Longwave radiation $R^2 = 0.77$.

### 4.1.2. Validation of Longwave Radiation

Figure 6 shows a comparison between the measured data and the estimated longwave radiation during a clear-sky day. When the SVF was high, the portion of the sky covered by clouds and composed of a single surface element (low-rise building, park, river, etc.) is overestimated. However, sites with low SVF and complex urban structures are underestimated at night. For example, in the commercial areas, narrow alleys, and residential areas, there are differences of approximately $-48$, $-31$, and $-31$ W/m$^2$ between the measured and estimated data.

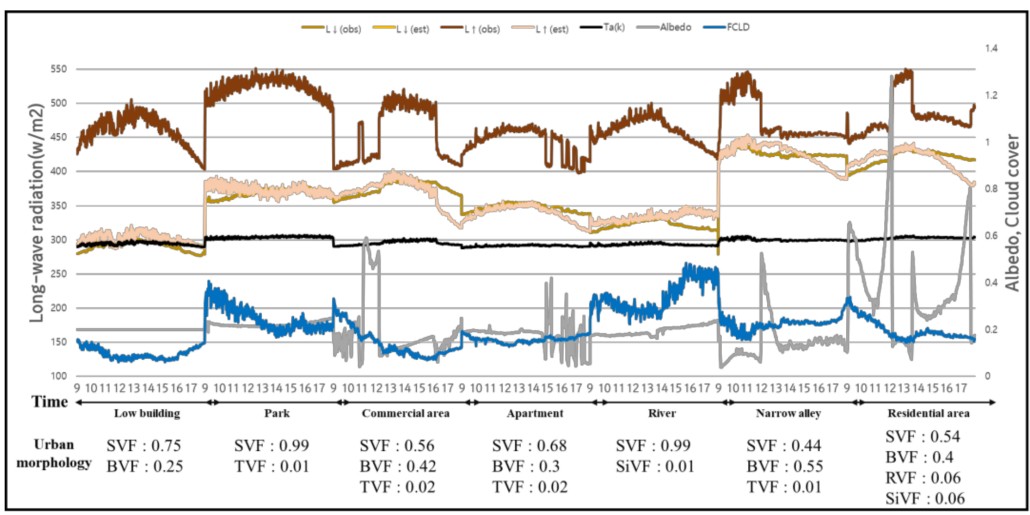

**Figure 6.** Analysis of incoming longwave radiation by comparison with meteorology data, albedo, and urban morphology from 7 sites. Cloud cover was calculated using air temperature (Ta) and relative humidity (RH).

Figure 6 shows a negative correlation between the albedo and outgoing longwave radiation. In addition, the rapidly changing albedo was greatly influenced by the presence or absence of shadows. Therefore, we can more accurately estimate the longwave radiation when considering the albedo value which changes with the altitude of the sun.

We estimated the longwave radiation by applying the heating coefficient value using the simplified surface energy balance equation. Because the sensible heat value is established based on the empirical formula of surface moisture and soil properties according to the temperature value, the accuracy is higher in dry weather [31,43]. By comparing the measured data, the longwave radiation of most areas decreased as cloud faction increased. However, there is a negative correlation between cloud faction and longwave radiation at parks and rivers area.

*4.2. Comparison of Estimated MRT with Other Models*

The comparison was focused on the simulation model (RayMan, SOLWEIG, and ENVI-met), which have been widely used in previous [44–46]. Based on the RMSE and index of agreement values (7.98 °C, 5.02 °C, 6.92 °C, 4.25 °C, and 0.92, 0.92, 0.89, 0.97, respectively), the MRT estimation method using deep learning was more accurate than other models [46]. RMSE in shortwave radiation is larger than that of other models on account of the amount of shortwave radiation transmitted through the trees and the error in estimating SVF. However, these factors have an insignificant effect on shortwave radiation (difference: 0.2 W/m$^2$). Although, the method proposed in this study overestimated shadow effect by ~46 W/m$^2$ (see Appendix C) and underestimated the short-wave radiation in sunlit area, this method estimated tends to be similar to other models.

In contrast, longwave radiation estimated by MRT_GSV showed the lowest RMSE values among the three models. These results show that the calculation formula which uses the absorption rate of shortwave radiation can accurately estimate longwave radiation in clear weather, except at night [32]. These errors indicate surface temperature estimation and shortwave reflected radiation calculations can be improved in the models because there is a rapid change depending on shortwave radiation variation.

The results of the comparison between the three models commonly used to calculate MRT at the city scale and the MRT_GSV method presented in this study show high overall accuracy with all models. In particular, this method can estimate the MRT through a 360° panorama image in an area which requires less input data than other models (only temperature and humidity). Therefore, this makes it easy to acquire data because it does not use 3D software programs and can construct data by taking 360° pictures.

A panorama image in a street canyon has the advantage of acquiring data from the 2.5D perspective and viewing many objects on one screen, including the point of view. Because 2D models, such as single layers and multilayers, are limited to the height of the same building in the horizontal space and cannot consider a 3D effect, the error is greater than 2.5D when calculating the VF in real space. Therefore, MRT evaluation on the 2.5D side can evaluate the MRT by three-dimensionally considering the complex urban spatial factors in the horizontal space. Although the accuracy may be lower than that of the 3D model, the results similar to those obtained using the software program evaluating MRT based on the 3D model can be confirmed, and the MRT estimated with image segmentation technique using deep learning can be quickly evaluated at the city scale.

*4.3. Comparison between LST and Estimated MRT*

We mapped MRT and LST, which have been used to evaluate the risk of heat-related diseases according to spatial characteristics in previous studies, to analyze spatial differences (Figure 7). For accurate analysis, we confirmed the: following (1) shortwave radiation under the effect of street morphologies and urban geometries at 10:00 am with LST; and (2) classified regions with high or low MRT based on a comparative analysis between LST and MRT (Figure 7, see Appendix D).

We assumed a clear-sky to analyze the effect of shortwave radiation according to the street canyons and only used panorama image data acquired during the summer period because leaves fall in other seasons and cause errors in the Tree View Factor (TVF) value.

The LST in Seoul was high (>30.2 °C) in areas with no obstacles and a large impervious surface, such as compact low-rise density buildings, compact mid-rise density buildings, and external parking lots. Low LST was distributed in the permeable surface (dense tree, low plants) and high-density building areas (approximately 17–26 °C) (see Appendix E). Similarly, the MRT of street canyons (>59.4 °C) was mainly distributed in open space areas (low plants, bare paved areas, etc.) and compact low-rise density buildings (SVF and BVF were approximately 0.6–1.0 and 0.35–0.5, respectively) or street canyons with West-East orientation (SVFs were ~0.3–0.55) (see Appendix E). SVF is the main cause of high MRT values in low plants with low LST. Otherwise, the lower MRT of street canyons (<47.6 °C) was mainly distributed in high buildings (compact mid-rise density and high-density

buildings) or tree coverage areas (BVF and TVF were ~0.4–0.6 and 0.6–0.99, respectively) (see Appendix E). In particular, the high land surface temperature compact mid-rise density building area showed a low MRT value, and the low plant's area showed a high MRT value (see Appendix E). Therefore, because the results of thermal stress analysis through LCZ classification differ between urban geometries and MRT, heat and temperature must be considered in thermal vulnerability analysis or health impact analysis [47,48].

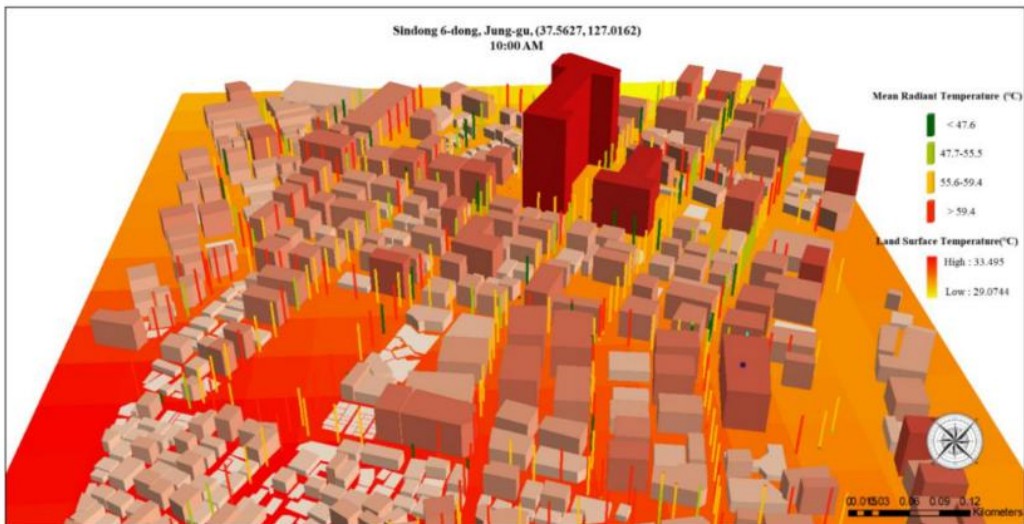

**Figure 7.** Comparison analysis between Land Surface Temperature and Mean Radiant Temperature at Sindong 6-dong, Jung-gu, Seoul which had a compact high and mid-density building. Bar graph mean MRT. We classified the MRT value into 4 classes based on the analysis result between MRT and heat risk in the previous study [3]. And surface mean LST using Landsat 8 images.

From these results, we can confirm the following: (1) TVF has a negative relationship with SVF, and the difference in the amount of the direct shortwave radiation is about 421 W/m$^2$ depending on the presence or absence of a shadow; (2) streets with a west-east orientation receive higher shortwave radiation than the other street orientations because they become horizontal with the path of the sun and the shaded area decreases; (3) compact mid-rise density building areas with low SVF and high BVF have high LST (approximately 29–35 °C) and low MRT (<47.6 °C). Conversely, low plant areas have low LST (approximately 17–26 °C) and high MRT (>59.4 °C).

### 4.4. Limitations and Future Developments

Accurate estimation of shade and surface temperature is the most important factor for calculating MRT. Although we calculated the solar irradiation by considering the shaded area ratio using the method from allen et al. [39], we needed height data to accurately estimate the proportion of the shaded area. Meanwhile, in the field of data science, a technique for estimating the height of a building in an image is in progress. It could be estimated with a more detailed surface temperature in the image by including an energy balance model, which would be part of future research. In addition, SVF, an important factor in estimating the shortwave radiation, used images taken from roads, not pedestrian paths. As the value of SVF varies greatly depending on the point where it is photographed, it is difficult to represent the thermal comfort of pedestrians with images photographed on the roadway. However, since we can produce 360-degree panorama images at any point through the street view app, it is easier to construct spatial data than other models (ENVI-met, SOLWEIG, etc.) and can be easily accessed. Therefore, the MRT can be estimated through a 360-degree image anywhere a user prefers.

In this study, the view factor was calculated by defining the ratio of urban elements visible at the point where a person is located. In addition, the *Fsw* used to estimate the long-

wave radiation was obtained from regression analysis with daily measurements, which cannot reflect the change during the day. However, by comparing the error between the estimated and measured values of long wave radiation affected by the view factor and *Fsw* with other models, it was confirmed that the availability of this method was high. Even though, based on the data from seven measurements in Seoul, we confirmed the potential of the proposed method, this is only possible in a clear-sky day. Thus, it is difficult to apply in regions with cloudy weather. However, when there is meteorological input data such as turbidity and atmospheric pressure, it can be used in all regions by using the method proposed in [19,49]. Finally, the meteorological data required to estimate MRT was used as AWS data. This cannot be reflected at all point locations in Seoul. Hence, slight errors may occur in the result of the estimated MRT mapping of Seoul city. However, with the recent increase in the field of citizen science, this limitation can be developed using wide-range and high-resolution citizen data in future research.

## 5. Conclusions

This study focused on: (1) how to improve the calculation of the MRT with shortwave and longwave radiation at the street level using publicly available panorama images; (2) investigating the effects of street canyon geometries (street orientation) and morphologies (sky view factor and view factors) on street-level shortwave radiation; and (3) analyzing the relationship between LST and MRT. Our developed method was verified using field measurements in various regions (residential areas, high-density buildings, parks, and open spaces) and we demonstrated that the clear-sky solar radiance of street canyons accurately captures the diurnal cycle in high-density environments ($R^2 = 0.97$). Although differences between the observed longwave radiation and estimated longwave radiation were found when the albedo changed with the altitude of the sun, which was calculated using a simplified energy balance model, we estimated a high accuracy ($R^2 = 0.77$). In addition, we found that the high MRT of street canyons ($\geq 59.4$ °C) is distributed in open space areas and compact low-rise density buildings (SVF $\geq$ 0.6 or 0.3–0.55 in a west-east street orientation, BVF: 0.35–0.5). But high-density buildings (BVF > 0.4) or high-density tree areas (TVF > 0.6) showed low MRT ($\leq 47.6$ °C). Finally, as a result of the comparison between MRT and LST, there was a difference between MRT and LST in low tree density or low-rise high-density building areas.

Generally, sites with high LST and MRT were distributed in similar spaces; however, low growing plants with low LST values as permeable layers showed high MRT values with high SVF (>0.95), and in compact low-rise density buildings with high LST values, indicate low MRT value do to surrounding obstacles (BVF and TVF were ~0.4–0.6, 0.6–0.99). The spatial variability of street-level MRT is closely related to the high tree cover. A lower MRT and high TVF occur in streets orientated north-south during the daytime. In particular, shortwave radiation has a large impact on the shaded area at the street level. The MRT estimation method presented in this study can be calculated from location, date, temperature, and humidity data. In particular, spatial data can calculate SVF and VF us-ing deep learning techniques based on panoramic images provided by Google Street View or directly captured panoramic images [24]. These data can provide a low-cost and effective streetscape mapping approach for urban areas. This method can also detect the presence of shadows through location and date data, and calculate radiant energy using temperature and humidity data. In addition, since temperature and humidity data can be used in urban spaces within a 4km buffer based on public data, it is expected that the methodology proposed will be highly applicable in urban street canyons where spatial data or climate measurement data is insufficient. The resulting maps of street-level shortwave radiation provide crucial datasets for studying the spatiotemporal variabilities of street-level MRT and understanding the interactions between shortwave radiation and human health at the street level. It also helps to provide more accurate MRT to city planners to plan green infrastructure implementation and mitigate the heat wave impacts on health in urban areas.

**Author Contributions:** All authors made substantial contributions to conception and design of the study. Conceptualization: software, methodology, writing, E.-S.K.; visualization, validation, resources, S.-H.Y.; formal analysis, editing the manuscript, C.-Y.P.; data curation, conceptualization, editing the manuscript, H.-K.H.; supervision, guided the research, reviewed the manuscript, D.-K.L. All authors have read and agreed to the published version of the manuscript.

**Funding:** This work was conducted with the support of the Korea Environment Industry and Technology Institute (KEITI) through its Urban Ecological Health Promotion Technology Development Project and funded by the Korea Ministry of Environment (MOE) (2020002770003).

**Institutional Review Board Statement:** Not applicable.

**Informed Consent Statement:** Not applicable.

**Data Availability Statement:** For this analysis we used publicly available data: LANDSAT 8 data (https://www.usgs.gov accessed on 9 September 2021). Satellite image acquired on 19 Jun 2018 (OLI & TIRS). To get google street view image data, we used open-source code provided by matlab (https://kr.mathworks.com/matlabcentral/fileexchange/50187-get_google_streetview-loc_v-varargin accessed on 9 September 2021). Finally, we used climate data (https://data.kma.go.kr/cmmn/main.do accessed on 9 September 2021).

**Conflicts of Interest:** The authors declare no conflict of interest.

## Appendix A

Instrument specifications for the measuring equipment. CNR4 was prepared for assessing the mean radiant temperature, shortwave radiation, and longwave radiation. Climate measurements employed a weather station equipped with air temperature, relative humidity, and wind speed sensors at 1.2 m above ground. The accuracy of the equipment is shown in Table A1. Data from all sensors were registered by a logger and recorded every 1 min.

**Table A1.** Parameters list measured and information of measurement sensor.

| Measured | Sensor | Unit | Accuracy |
|---|---|---|---|
| Air temperature | S-thb-m002 | °C | ±0.21 °C |
| Relative Humidity | | % | ±2.5% |
| Wind speed | S-wcf-m003 | m/s | ±1.1 m/s |
| Shortwave radiation | CNR4 | $W/m^2$ | ±10% |

## Appendix B

In this study, a resnet-18 convolutional neural network was used in the MATLAB program for image classification. Aver IoU means, on average, whether the object location of all classes was accurately detected. Additionally, per class IoU means whether the object location is accurately detected according to each class (sky, tree, road, sidewalk, building). The deep learning method for image classification is as follows:

(1) Download the urban image dataset, (2) encapsulate the pixel label data and group the 5 classes using the label ID to a class name mapping, (3) split images evenly into 60%, 20%, 20% for training, validation and testing, respectively, (4) create a DeepLab v3+ network based on ResNet-18 in MATLAB, (5) start training using train network and using Deep Learning Toolbox in MATLAB, (6) test network on images.

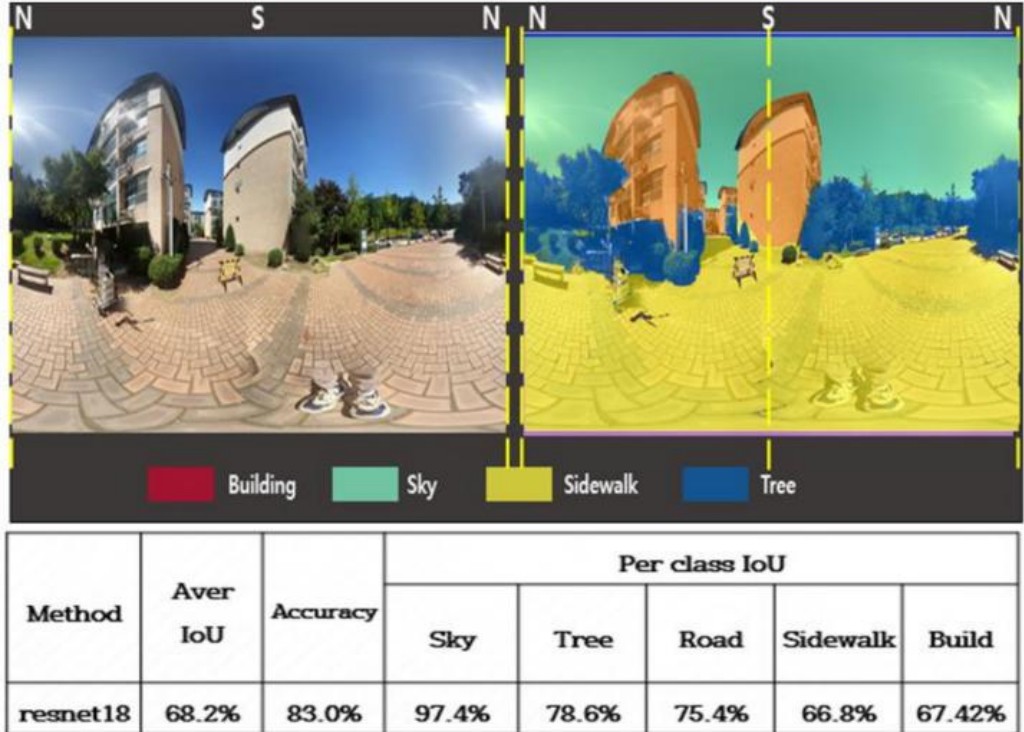

| Method | Aver IoU | Accuracy | Per class IoU | | | | |
|---|---|---|---|---|---|---|---|
| | | | Sky | Tree | Road | Sidewalk | Build |
| resnet18 | 68.2% | 83.0% | 97.4% | 78.6% | 75.4% | 66.8% | 67.42% |

**Figure A1.** Workflow procedure for image segmentation to calculate VF using deep learning. ResNet 18 converged faster than other methods and was comparably accurate [30].

## Appendix C

Validation of short-wave and long-wave radiation at field measurements in the clear-sky. Figure A2 shows that the clear-sky solar radiance of street canyons for a day in high-density and low-building environments. In order to increase the reliability of the method proposed in this study, it was verified in various places.

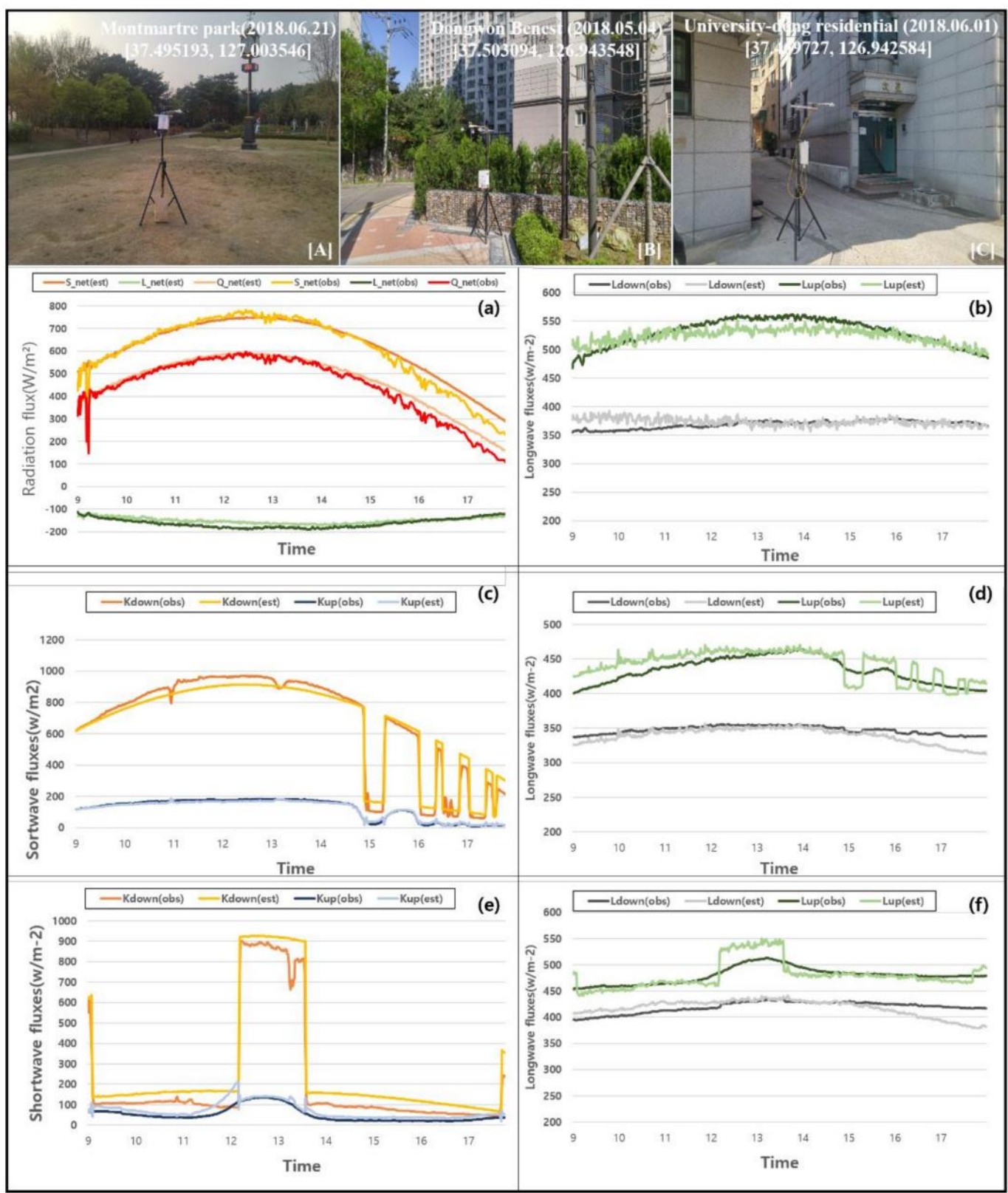

**Figure A2.** The clear-sky solar radiance of street canyons for a day in high-density and low-building environments in various places. (**A**) Montmartre park (**B**) Dongwon Benest (**C**) University. Comparison between measurement and estimated radiation (**a**)–(**f**).

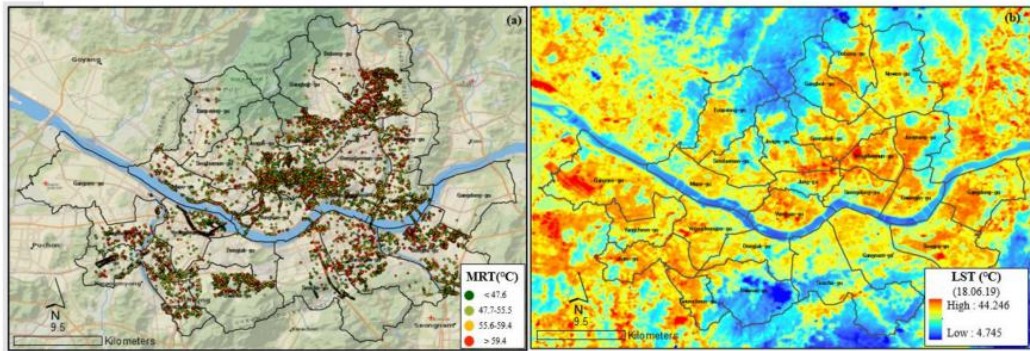

**Figure A3.** The clear-sky solar radiance of street canyons for a day in high-density and low-building environments in various places. (**D**) University-dong residential on 10 May 2018 and (**E**) University-dong residential on 4 June 2018. Comparison between measurement and estimated radiation (**g**)–(**j**).

**Appendix D**

MRT (°C) for a total of 58,794 of street canyons in Seoul on 19 June 2018, at 10:00 a.m. (b) LST (°C) using Landsat 8 satellite imagery.

**Figure A4.** (**a**) Estimated MRT mapping using Google Street View in Seoul. (**b**) Land Surface Temperature map using Landsat 8 data in Seoul.

## Appendix E

Detailed information describing Figure 7 is presented in Table A2: Point location data for comparative analysis between LST and MRT. We focused on urban morphology and street orientation to analyze in which regions the MRT-LST difference is significant. Medium heat risk level was used to compare the degree of risk by urban area. The threshold value for the effect of MRT on human health was classified based on the results of previous studies (risk 1 < 47.6 °C, 47.6 °C < risk2 < 55.5 °C, 55.5 °C < risk 3 < 59.4 °C, risk4 > 59.4 °C), and LST was classified as risk1 <20.8 °C, 20.8 °C <risk2 < 24.7 °C, 24.7 °C < risk 3 < 30.2 °C, risk4 > 30.2 °C. Since LST is affected differently by urban area, the degree of impact on human health due to heat was identified by classifying it by the quantiles used in previous studies. As a result of the analysis, the CMDB with high LST showed a low MRT value, and the LP region with low LST showed a high MRT value. Where CLDB is compact low-rise density building, CMDB is compact mid-rise density building, DT is dense tree, LP is low plants and HDB is high density building.

**Table A2.** Urban morphology, MRT, LST data for each point location.

| | Urban Morphology | Location | Lat | Lon | LST | SVF | TVF | BVF | Street Orientation | MRT |
|---|---|---|---|---|---|---|---|---|---|---|
| High LST | compact low-rise density building | 205-422, Cheongnyangni-dong, Dongdaemun-gu | 37.5895 | 127.0414 | 31.627 | 0.75 | 0 | 0.25 | N-S | 66.1 |
| | | Anam-ro 24-gil, Jegi-dong, Dongdaemun-gu | 37.5882 | 127.0362 | 33.015 | 0.64 | 0.01 | 0.35 | E-W | 62.1 |
| | | Changsin 1-dong, Jongro-gu | 37.5718 | 127.0139 | 32.141 | 0.41 | 0 | 0.59 | N-S | 61.3 |
| | | 977-18, Bangbae-dong, Seocho-gu | 37.4815 | 126.9923 | 32.542 | 0.6 | 0.02 | 0.38 | E-W | 61.8 |
| | | Munrae-dong 4-ga, Yeongdeungpo-gu | 37.5147 | 126.8906 | 33.771 | 0.7 | 0 | 0.3 | E-W | 65.4 |
| | bare paved area | Suseo station parking lot | 37.4854 | 127.1056 | 34.783 | | | | - | |
| | | 735, Suseo-dong, Gangnam-gu | 37.4878 | 127.0998 | 35.232 | | | | | |
| | | Ilwonbon-dong, Gangnam-gu | 37.4874 | 127.0801 | 35.547 | | | | | |
| | compact mid-rise density building | 279-47 Sangdo 4-dong, Dongjak-gu | 37.4957 | 126.9374 | 29.341 | 0.42 | 0 | 0.58 | NE-SW | 43.9 |
| | | 41-5, Hwayang-dong, Gwangjin-gu | 37.5451 | 127.0666 | 29.997 | 0.38 | 0 | 0.62 | N-S | 42.2 |
| | | 254-239, Daehak-dong, Gwanak-gu | 37.4649 | 126.9359 | 31.011 | 0.3 | 0 | 0.7 | E-W | 40.1 |
| | | 9-34, Suyu3-dong, Gangbuk-gu | 37.6383 | 127.0205 | 30.014 | 0.55 | 0 | 0.45 | E-W | 44.5 |
| Low LST | dense tree | Nakseongdae park | 37.4719 | 126.9599 | 18.354 | 0.03 | 0.97 | 0 | E-W | 33.8 |
| | | Janggunbong Sports Park | 37.4787 | 126.9384 | 18.997 | 0.01 | 0.99 | 0 | E-W | 35.1 |
| | | 44-3 Ogeum-dong, Songpa-gu | 37.5051 | 127.1277 | 19.584 | 0.3 | 0.6 | 0.1 | NE-SW | 32.4 |
| | low plants | Montmartre park | 37.4954 | 127.0038 | 21.711 | 0.99 | 0 | 0.01 | NE-SW | 70.2 |
| | | Yeouido hangang park | 37.5293 | 126.9326 | 22.667 | 0.96 | 0 | 0.04 | N-S | 69.1 |
| | | pyeonghwaui park | 37.5618 | 126.8907 | 23.421 | 0.95 | 0 | 0.05 | E-S | 68.8 |
| | high density building | 460 Hongje-dong, Seodaemun-gu | 37.5854 | 126.9506 | 24.145 | 0.55 | 0.03 | 0.42 | E-W | 47.1 |
| | | 140 Garak-dong, Songpa-gu | 37.4956 | 127.1278 | 25.245 | 0.4 | 0 | 0.6 | N-S | 43.8 |
| | | 467-7 Dogok-dong, Gangnam-gu | 37.4882 | 127.0519 | 26.114 | 0.42 | 0.21 | 0.37 | N-S | 44.2 |
| | | 27-45 Sangdo 2-dong, Dongjak-gu | 37.5043 | 126.9433 | 25.773 | 0.52 | 0.02 | 0.46 | E-W | 46.8 |

**Table A2.** *Cont.*

|  | Urban Morphology | Location | Lat | Lon | LST | SVF | TVF | BVF | Street Orientation | MRT |
|---|---|---|---|---|---|---|---|---|---|---|
| | | | | | **Medium of heat risk level** | | | | | |
| **No** | **Urban morphology** | **Land Surface Temperature (mean/sd)** | | | | **Mean Radiant Temperature (mean/sd)** | | | | |
| **1** | CLDB | 3.871/0.336 | | | | 4/0.12 | | | | |
| **2** | CMDB | 3.38/0.486 | | | | 1.157/0.364 | | | | |
| **3** | DT | 1/0.05 | | | | 1/0.04 | | | | |
| **4** | HDB | 2.501/0.5 | | | | 1.352/0.478 | | | | |
| **5** | LP | 1.75/0.434 | | | | 4/0.23 | | | | |

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
