# Peer review of "Estimation of Mean Radiant Temperature in Urban Canyons Using Google Street View: A Case Study on Seoul"

_remotesensing, doi:10.3390/rs14020260_

Round 1

Reviewer 1 Report

Dear authors,

thank you for your contribution in this topic.

I do have some comments, but basicly I'm not sure if I have understood the presentation of the research. Here are some comments:

The abbreviation first in use has to be with whole words and abbreviation in (); Replace in the line 13 and 14 (MRT)

A lot of grammar mistakes of citations, see Authors instructions, correct the English language.

Google street view is not a method, just a tool, I hope. If it is a tool, then the explanation of the research, method and all steps has to be redefined. If it is meant as a method, I do not agree with the research, as the GSV is a process of images done just once per year (in some cases). Here all the method and tools need more explanation. If I have understood it wrong, it means that it has to be rewritten. In all, not enough good to understand.  

In the chapter 4 results it is clear to me that the process scan be a good research, but if explaining through GSV as a method it loos the research meaning.

In conclusion, the first part is the abstract of the research, insert it in the chapter of results.

Line 416-423 has to be cited: Fang-Ying Gong, Zhao-Cheng Zeng, Edward Ng, Leslie K. Norford. "Spatiotemporal patterns of street-level solar radiation estimated using Google Street View in a high[1]density urban environment" , Building and Environment, 201.

Reference list to be corrected: name of jurnals (papers) is in italic, year bold, ISBN is not the type to add to the reference, title of the book in italic, etc., see Authors instructions.

Reviewer 2 Report

The manuscript entitled “Estimation of mean radiant temperature in urban canyons using Google Street View” proposed a method to estimate MRT based on street view images. The authors validated the estimations against in situ measurements. This work could be meaningful for urban planning. However, most of the methods used in this study are rather unclear (e.g., measurements, model simulations, and deep learning model). It is also unclear how users can apply the proposed method without measurement data. Here are my major concerns and some minor comments:

Major concerns:

Figure 2: I do not think Fig. 2 is necessary here given that it is simply showing the structure of the paper.

Figure 4: Unclear what “Aver IoU” and “Per class IoU” are. The authors also need to elaborate on the DL methods used here.

Figure 5: Variables in the table are unclear. Are X-axis and Y-axis latitude and longitude??

Lines 171-172: Please elaborate on how the shadow was determined.

Equation 1: Please define FCLD first. In Lines 189-193: it is unclear how ~0.75 was derived from elevation. Please also thoroughly check all equations as there are many undefined variables.

Line 224: Please explain what the heating coefficient is. Is it a conversion factor from incoming shortwave radiation to longwave? Did the authors consider multi-reflections between canyon surfaces?

Equation 10: How did the authors determine the surface temperature T?

Section 4.1: Details of measurements are needed. Currently, only site images are shown in appendix A, but no information regarding the quality of these measurements and types of instruments is mentioned. The authors compared the VF-based estimation against measurements, but it is unclear whether these measurements were conducted at the sites where google street view images were taken.

Lines 265-272: The authors concluded that the estimations are “satisfactory”. However, this is not the case, especially for net longwave radiation and MRT. In fact, MRT error can be up to ~10 °C (e.g., at around 11-12 am).

Figure 9 and Appendix A3: It is extremely hard to find out whether the shown MRT and LST patterns are consistent with each other. In addition, were LST and MRT on the same day?

Section 4.3: It is hard to judge whether this part is reasonable or not without any details of model simulations (e.g., model setups, validation, accuracy assessments).

Minor comments:

Line 24: minus 0.3 for SVF?

Lines 39-42: The authors mentioned “a previous study” but cited four references in Line 42.

Line 199: Please explain why a threshold of 4 km was selected here.

Line 180: Should be Table 2.

Lines 322-323: “difference … is significantly dependent on the presence or absence of a shadow (approximately 421 w/m²)” – what is this 421? Difference?

Reviewer 3 Report

Kim et al. submitted the article Estimation of mean radiant temperature in urban canyons 2 using Google Street View to mdpi remote sensing. The authors developed a new method to improve the estimation of MRT in urban areas using Google Street View data. The method is more accurate as it gives a better estimation of radiation in streets as it can better capture wall radiation, shadow path meaning 3S effect that satellites won't be able to capture correctly. Moreover, it provide higher resolution that available data. The study is presented in a logical way and investigates complementary aspects of the results. Conclusion is supported by convincing Figures, method description and a case study. There are a couple of sentences that need to be revised for better clarity,, all parameters in equations properly described, Affiliations added to all authors ect … Please look at the specific comments for detail.

Questions

Q1

Is the novelty of your method is it the simplicity of implementation as you use GSV data or the fact that you use 3D radiation input ? instead of a 2D map that only considers incoming radiation from the sun and not the reflected ones/building heat emission etc ... ?

Q2

line 253 discusses the discrepancy between the model and estimate LW. Is the surface albedo accurate enough ? pavement may have slightly different albedo and absorption coefficient. In what sense this can impact your results  as you are using empirical homogeneous coefficients .

Q3

Looking at Figure 6, what could explain those high frequency SW observations between 12-13 that are also captured by MRT (obs) but not modelled properly.

Similar happen 16-17 and 9-10am. It seems to be located just after a large change of radiation may be connected with building orientation ? Is that linked with material heterogeneity ?

Q7 About Figure 7

Could you explain why LW presents larger uncertainty ? What explain the type of graph observed Fig7 (right) ?

Title

Precise the location of your study case

Estimation of mean radiant temperature in urban canyons using Google Street View

-->

Estimation of mean radiant temperature in urban canyons using Google Street View: A case study on Seoul

Tables

Table 1

Cloud cover --> Cloud cover (in %)

Specific comments

Line 5-9 Affiliation is missing ... should be lab/address/country

Line 14, define MRT before using it = use mean radiant temperature (MRT) here then no need Line 15

Line 18-20 Unclear sentence

You mean the method was validated ?at street level with a RMSE of 0.97 an d0.77. You need to precise the exact statistical method you use to validate model vs. observation.

Line 24

SVF ??

Define it before using it

Line 36

As fu-

ture scenarios of heat-related diseases

-->

As future climate change scenarios of heat-related diseases

Line 39 You need to summarize their general conclusion or say that using LST satellite measurement is for example an interesting approach that provides accurate results compared to observations. Although, this data can be used to map potential hazardous areas in cities affected by heat waves ...or provide support for better green infrastructure to mitigate heat waves ....

Line 39

previous study

-->

several studies

Line 43 You can also indicate a range of scale (25km ??)

spatial resolution

-->

coarse spatial resolution

Line 60

, the formula

-->

, the MRT formula

Line 75

unique method

-->

new method

Line 89

Move the Fig1 as close as possible as the place you call it so on top of page 3

Line 97

urbanization.

-->

urbanization and lack of green infrastrures that could mitigate.absorb the heat.

Line 100 ?

is predicted

-->

is high and predicted

Line 101

a representative area

-->

a representative highly urbanized area

Line 108 ?

2.2. Data collectionrea

-->

2.2. Data collection area

Line 136 you mean using the He et al. approach ??

Line 149

Sun

-->

sun

Line 155 correct the author name

MOTRONL.BARAD.[34] ??

Line 184

0.75Extraterrestria

-->

0.75 Extraterrestria

Line 186

after introduction equation 1 , you need to define each variable, what is Ra? and FCLD ? and Ig ??

Line 198

Ideam as preiously Sdiff ??

Line 204, 205, and 206

D? and f ?

Define Eq 9 and 10 parameters

FSw ?? Sdiff ?? VFmaterials .. etc ... all variables need to be defined after being introduced.

Line 268-271

Unclear sentence. You mean that SW and LW differ but present similar trends ?

Line 276

PLease move the Figure 8 there to be just after the description of Figure 8.

Line 282

Fig. 8 shows a negative correlation between the albedo and longwave radiation. ?

Do you mean for each type of area except the river ? Could you indicate the value of the correlation factor ?

Line 290-291

longwave radiation increases when the daytime cloud fraction (FCLD) ??

Could you provide some value of variability of LW vs daytime cloud fraction to support that argument.

Line 303

Move Figure 9 at the top of the page as it is close to the Figure citation in the text.

Line 304

Define TVF acronym before using it

Line 307

Is permeability the only effect? This of course suppose that the soil may be presenting more moisture due to its porosity but trees/plants also have a cooling effect due to shade ... and effect of roots/leaves on moisture.

Line 341

difference

-->

bias/RMSE/error

Line 344 and 345

w/m2

-->

W/m2

Line 349

longwave radiation

-->

longwave radiation estimated by MRT_GSV

Line 386-387

possibility for the

-->

potential of the

Line 392

. hence

-->

. Hence,

Line 396

to calculate the MRT

-->

to improve the calculation the MRT

Line 412 strange sentence

the MRT value was low due

-->

indicate low MRT value do to

Line 423

It also helps to provide more accurate MRT to city planners to plan green infrastructure implementation and mitigate heat wave impact on health in urban areas .... or provide more accurate data to heatmap data for emergency.

Round 2

Reviewer 1 Report

Dear authors,

thank you. You have improved the paper and I agree on all your comments,

with kind regards

Author Response

Thanks for reviewing our paper. We have added and corrected the description of the sentence structure and parameter variables in our manuscript. And the reason why the methodology is useful and the input data needed to use it and how to use it are briefly added in the conclusion section.

Reviewer 2 Report

Thanks for the effort. The manuscript now becomes clearer. But I still have some additional comments:

Lines 237-239: Following my previous comment #6, did the authors calculate the view factors between each pair of urban elements (e.g., tree to wall)? The authors used a constant heat coefficient for each surface, while the conversion should exhibit diurnal variation. Will the use of this constant coefficient lead to any uncertainties?

Authors’ response to my comment #7: Sorry, but I do not understand what you meant by “the availability of model was increased by using Fsw as the limit of surface temperature”.

Authors’ response to my comment #7: “In this study, the surface temperature (T) was used as the air temperature.” – this is a rather rough estimation, which might lead to very high uncertainties. Surface temperature can easily be much higher than air temperature (e.g., over pavements). What are the “unstable conditions”? Do you mean unstable boundary layer conditions?

Authors’ response to my comment #10: Thanks for the explanation, but the presentation of this figure is still unclear to the readers. Perhaps some statistics might help.

The authors did not respond to my previous comment: “It is also unclear how users can apply the proposed method without measurement data.”

Author Response

Thanks for reviewing our paper. We have added and corrected the description of the sentence structure and parameter variables in our manuscript. And the reason why the methodology is useful and the input data needed to use it and how to use it are briefly added in the conclusion section. And we added comments on heat coefficient and surface temperature. Additionally, the table has been added to the appendix to help users understand the difference analysis between LST and MRT.

Round 3

Reviewer 2 Report

I have no additional comments.